# Effectiveness and Safety of Ceftriaxone Compared to Standard of Care for Treatment of Bloodstream Infections Due to Methicillin-Susceptible *Staphylococcus aureus*: A Systematic Review and Meta-Analysis

**DOI:** 10.3390/antibiotics11030375

**Published:** 2022-03-10

**Authors:** Yazed Saleh Alsowaida, Gregorio Benitez, Khalid Bin Saleh, Thamer A. Almangour, Fadi Shehadeh, Eleftherios Mylonakis

**Affiliations:** 1Division of Infectious Diseases, Alpert Medical School, Brown University, Providence, RI 02903, USA; fadi_shehadeh@brown.edu; 2Division of Infectious Diseases, Rhode Island Hospital, Providence, RI 02903, USA; gbenitez@lifespan.org; 3Department of Clinical Pharmacy, College of Pharmacy, Hail University, P.O. Box 6166, Hail 81442, Saudi Arabia; 4College of Pharmacy, King Saud bin Abdulaziz University for Health Sciences, King Abdullah International Medical Research Center, Pharmaceutical Care Department, King Abdulaziz Medical City, P.O. Box 3660, Riyadh 11481, Saudi Arabia; binsalehkh@gmail.com; 5Department of Clinical Pharmacy, College of Pharmacy, King Saud University, P.O. Box 2457, Riyadh 11451, Saudi Arabia; talmangour@ksu.edu.sa; 6School of Electrical and Computer Engineering, National Technical University of Athens, 10682 Athens, Greece

**Keywords:** ceftriaxone, bacteremia, bloodstream infection, methicillin-susceptible *Staphylococcus aureus*, MSSA, outpatient parenteral antimicrobial therapy

## Abstract

(1) Background: Ceftriaxone is a potential alternative for the treatment of methicillin-susceptible *Staphylococcus aureus* (MSSA) bloodstream infections (BSIs) in acute care and outpatient parenteral antimicrobial therapy (OPAT) settings. We evaluated the effectiveness and safety of ceftriaxone for the treatment of MSSA BSIs. (2) Method: We searched PubMed, Embase, and Cochrane Library from their inception to October 30th 2021. Our outcomes included clinical cure, microbiological cure, 30- and 90-day mortality, 90-day hospital readmission, and adverse drug reactions (ADRs). We compared ceftriaxone against standard of care (SOC) therapy. We used the random-effects model for the meta-analysis, and our estimated effects were reported as odds ratios (ORs) with 95% confidence intervals (CI). (3) Results: Twelve retrospective cohort studies were included, comprising 1037 patients in the ceftriaxone arms and 2088 patients in the SOC arms. The clinical cure rate of the ceftriaxone regimen was not statistically different from SOC: OR 0.65 (95% CI: 0.29–1.45). Ceftriaxone was also not statistically different from SOC in microbiological cure: OR 1.48 (95% CI: 0.29–7.51); 30-day mortality: OR 0.79 (95% CI: 0.14–4.65); 90-day mortality: OR 0.82 (95% CI: 0.38–1.80); 90-day hospital readmission: OR 1.20 (95% CI: 0.92–1.56); and ADRs: OR 0.92 (95% CI: 0.39–2.18). (4) Conclusion: Ceftriaxone could provide an alternative for the treatment of MSSA BSIs in acute care and OPAT settings (except in patients whose BSIs were due to infective endocarditis).

## 1. Introduction

*Staphylococcus aureus* is one of the most common pathogenic Gram-positive bacteria, and it causes a wide range of community-acquired and hospital-acquired infections [1,2]. *S. aureus* expresses several virulence mechanisms and can cause serious infections associated with mortality that can be as high as 22–48% [3,4]. In the United States (U.S.), *S. aureus* accounts for 23% of bloodstream infections (BSIs) [1,5]. Patient groups at increased risk for *S. aureus* BSIs include individuals aged 70 years and older, individuals with HIV, those who inject drugs, and patients on hemodialysis [3].

The usual length of treatment for methicillin-susceptible *S. aureus* (MSSA) BSI is a minimum of 14 days for an uncomplicated infection and up to 4–6 weeks for complicated infection [1,3]. Specifically, parenteral anti-staphylococcal penicillin and first-generation cephalosporin antibiotics are the mainstay of therapy for MSSA BSI. While nafcillin, oxacillin, and cefazolin are the standard of care (SOC) for MSSA BSI, ceftriaxone can be administered once daily compared to nafcillin or oxacillin (6 times daily) and cefazolin (3 times daily), and it can be given peripherally by intravenous (IV) bolus injection [6]. Thus, ceftriaxone is convenient to use, especially in outpatient parenteral antimicrobial therapy (OPAT) settings without hospitalization [7]. Delivery options in OPAT settings include home-based, infusion center-based, and skilled nursing facility-based.

Several studies have evaluated ceftriaxone for the treatment of MSSA BSIs [8,9,10]. However, these studies have conflicting results and limited sample sizes; thus, robust evidence is inconclusive. A recent systematic review and meta-analysis and another systematic review evaluated ceftriaxone use for MSSA infections [11,12]. However, these studies evaluated mixed infections and did not include all of the available studies. The objective of this study is to evaluate the evidence for the effectiveness and safety of ceftriaxone for MSSA BSI in acute care and OPAT settings qualitatively utilizing a systematic review and quantitatively by meta-analysis.

## 2. Materials and Methods

We followed the Preferred Reporting Items for Systematic Reviews and Meta-Analyses (PRISMA) and the Meta-Analyses of Observational Studies in Epidemiology (MOOSE) guidelines [13,14]. PRISMA and MOOSE checklists are available in Appendix A, respectively.

### 2.1. Literature Source

We performed a systematic search in PubMed, Embase, and Cochrane Library from their inception to 30 October 2021. We executed the search with the following concepts: ceftriaxone, BSI, and MSSA. The complete search strategy is available in Appendix A. In addition, we manually searched the citations of key studies.

### 2.2. PICOS Criteria and Study Selection

We searched using the following modified Population, Intervention, Comparator, Outcomes, and Studies (PICOS) criteria [15]. (1) Population: adult patients aged 18 years and older with MSSA BSIs; (2) intervention: ceftriaxone antibiotic; (3) comparator: anti-staphylococcal antibiotics (nafcillin, oxacillin, cloxacillin, and cefazolin); (4) outcomes: clinical cure; microbiological cure; 30-day mortality; 90-day mortality; 90-day hospital readmission; and adverse drug reactions (ADRs); (5) studies: randomized controlled trials and observational studies. Study investigators (Y.S.A. and G.B.) reviewed the titles and abstracts of all identified studies, and potentially eligible studies were retrieved for full-text review. Studies that met the PICOS criteria were included in the systematic review and meta-analysis. We excluded case reports, case series, editorials, commentaries, letters, studies with BSIs caused by pathogens other than MSSA, and publications in a language other than English.

### 2.3. Data Extraction

Two reviewers (Y.S.A. and G.B.) independently screened the titles and abstracts of all the identified studies in the databases for eligibility. We resolved discrepancies through discussion and consensus; unresolved matters were reviewed by a third person (F.S.). We extracted the following information for the included studies using a predeveloped worksheet: author and publication date, location and date of the study, sample size, sources of BSIs, and antibiotic regimens used. For studies that reported clinical/microbiological failure, we calculated the clinical/microbiological cure as the total number of patients minus the number of patients who failed.

### 2.4. Quality Assessment

The quality evaluation of the included studies was performed using the star system of the Newcastle–Ottawa Scale (NOS) for observational studies [16]. The studies were categorized based on their individual score: low quality (1–3 stars), medium quality (4–6 stars), and high quality (7–9 stars).

### 2.5. Summary Measures and Statistical Analysis

The meta-analytic computations were performed using STATA 17, StataCorp LLC, College Station, TX, USA. The pooled estimated effects were reported as odds ratios (ORs) with 95% confidence intervals (CI). We used the restricted maximum likelihood random-effects model to account for heterogeneity in the effect sizes of the studies [17]. We used Cochrane’s Q to calculate the heterogeneity by the weighted sum of squares, and the I^2^ statistic was used to express the percentage of variation due to heterogeneity [18]. Statistically significant heterogeneity was set at a threshold of *p* < 0.05 for Cochrane’s Q statistic and > 30% for I^2^ statistic. We assessed publication bias with funnel plots of standard error against the estimated effect. We used the Egger linear regression test method to evaluate the asymmetry of the funnel plot [19].

### 2.6. Subgroup Analysis

To evaluate the clinical cure rates based on the acuity level of the patients, we performed a subgroup analysis based on the treatment setting of either acute care or OPAT.

## 3. Results

### 3.1. Study Selection

The initial search yielded 54 relevant abstracts in PubMed, 160 abstracts in Embase, and 0 abstracts in Cochrane Library. There were three abstracts identified from the review of citations and conferences. After removing duplicates, 50 abstracts were screened. We further excluded three abstracts that were deemed irrelevant based on the review of the titles. In total, 47 abstracts were reviewed. After removing 26 irrelevant studies and 1 in-vitro study, 20 articles underwent a full-text review for eligibility. Based on the full-text review, eight articles were excluded and a total of 12 studies were included in the systematic review, while 11 studies were included in the meta-analysis (Figure 1).

Of note, in the retrospective cohort study by Paul et al. (2010) [20], ceftriaxone and cefotaxime were combined in a single arm. The sample size of this arm was 194 patients (ceftriaxone = 176 and cefotaxime = 18). We decided to include this study because cefotaxime is similar to ceftriaxone in pharmacologic class (a third-generation cephalosporin), has a similar spectrum of microbial activity, and only comprises 9.2% of the patient cohort. Moreover, Hamad et al. (2020) [9] and Hamad et al. (2021) [21] were conducted at the same hospital site but with different reported outcomes and conducted in different time periods. Hamad et al. (2020) [9] had a sample size of 243 patients and reported 90-day mortality, microbiological cure, and 90-day hospital readmission. In contrast, Hamad et al. (2021) [21] had a sample size of 1895 patients and reported 90-day hospital readmission only. Therefore, we excluded Hamad et al. (2020) [9] in our analysis of 90-day hospital readmission.

### 3.2. Systematic Review and Characteristics of the Included Studies

All the 12 studies included in the systematic review and meta-analysis were retrospective cohort studies [8,9,10,20,21,22,23,24,25,26,27,28]. Eleven of the included studies [8,9,10,20,21,22,23,24,25,26,28] were of high quality, and 1 study [27] was of medium quality (Appendix A). Eleven studies [8,9,10,21,22,23,24,25,26,27,28] were conducted in the U.S., and 1 study (Paul et al. [20]) was conducted in Israel. Notably, the study by Paul et al. was comparable in the design, the patient population included (age of the patients, treatment setting, patients’ comorbidities, and sources of BSIs), and the antibiotic treatment used to the rest of the studies. Seven of the included studies [8,10,20,22,23,25,26] were conducted in acute care settings, and five studies [9,21,24,27,28] were conducted in OPAT settings. The total number of patients included in the meta-analysis was 3125 patients, with 1037 patients in the ceftriaxone arms and 2088 patients in the SOC arms. There were a variety of suspected sources of MSSA BSIs among patients. Characteristics of the included studies are available in Table 1.

### 3.3. Meta-Analysis

#### 3.3.1. Clinical Cure

Seven studies [8,10,22,23,24,26,28] reported data on clinical outcomes and cure rate for patients with MSSA BSIs that included 235 patients in the ceftriaxone arms and 427 patients in SOC arms. The pooled clinical cure rate when using a ceftriaxone regimen was not statistically different from SOC: OR 0.65 (95% CI: 0.29–1.45; Figure 2A). In the subgroup analysis for acute care settings, five studies [8,10,22,23,26] were included that comprised 198 patients in the ceftriaxone arms and 364 patients in SOC arms. Similarly, the clinical cure when using a ceftriaxone regimen was not statistically different from SOC: OR 0.75 (95% CI: 0.30–1.91; Figure 2A).

#### 3.3.2. Microbiological Cure

Three studies [8,9,22] reported data on microbiological cures for patients with MSSA BSIs. The studies included 210 patients in the ceftriaxone arms and 169 patients in the SOC arms. The pooled microbiological cure rate when using a ceftriaxone regimen was not statistically different from SOC: OR 1.48 (95% CI: 0.29–7.51; Figure 2B).

#### 3.3.3. Mortality Due to Treatment Failure

Five studies [8,9,10,20,22] reported data on mortality secondary to the treatment failure when using a ceftriaxone regimen for the treatment of MSSA BSI. For the assessment of 30-day mortality, three studies [8,20,22] were included with 256 patients in the ceftriaxone arms and 205 patients in SOC arms. The pooled OR when using a ceftriaxone regimen was not statistically different from SOC: OR 0.79 (95% CI: 0.14–4.65; Figure 3A). Moreover, for the assessment of 90-day mortality, we included three studies [9,10,22] with 223 patients in the ceftriaxone arms and 184 patients in the SOC arms. In the assessment of 90-day mortality, the pooled OR when using a ceftriaxone regimen was not statistically different from SOC: OR 0.82 (95% CI: 0.38–1.80; Figure 3B).

#### 3.3.4. 90-day Hospital Readmission Due to Treatment Failure

Four studies [10,21,26,28] comprising 603 patients in the ceftriaxone arms and 1657 patients in the SOC arms reported data on hospital readmission due to treatment failure for patients with MSSA BSIs. The pooled OR when using a ceftriaxone regimen was not statistically different from SOC: OR 1.20 (95% CI: 0.92–1.56; Figure 4A).

#### 3.3.5. Adverse Drug Reactions

Five studies [9,10,22,26,28] comprising 333 patients in the ceftriaxone arms and 368 patients in the SOC arms reported data on ADRs for patients with MSSA BSIs. The rate of ADRs when using a ceftriaxone regimen was not statistically different from SOC: OR 0.92 (95% CI: 0.39–2.18; Figure 4B).

### 3.4. Publication Bias for the Clinical Cure

The funnel plot of standard error against the effect estimate reveals the overall symmetry of the included studies (Appendix A). Additionally, the Egger test did not reveal statistically significant publication bias (*p* = 0.77).

## 4. Discussion

Administration of ceftriaxone for MSSA BSIs is convenient because it is a once-daily antibiotic regimen [9]. In this systematic review and meta-analysis, we evaluated the use of ceftriaxone for the treatment of MSSA BSI. We evaluated clinical and microbiological cures, 30- and 90-day mortality, 90-day hospital readmission, and ADRs and found that ceftriaxone use was non-inferior to SOC in clinical effectiveness and safety outcomes that were evaluated. Our meta-analysis included five studies [8,10,22,23,26] that were conducted in acute medical care settings, and ceftriaxone maintained the clinical effectiveness for those patients based on our subgroup meta-analysis of acute care settings.

Methicillin-susceptible *S. aureus* BSIs can be associated with high morbidity and mortality, and thus using an effective antibiotic therapy is warranted [1]. Our findings suggest that ceftriaxone could provide an alternative treatment for MSSA BSI, and this is consistent with several studies. Barber et al. compared ceftriaxone to SOC for MSSA BSIs in acute medical care settings and found a similar clinical cure (ceftriaxone 78% vs. SOC 50%, *p* = 0.052) [8]. Mohamed et al. compared ceftriaxone to cefazolin for MSSA BSIs in a multicenter study and found a similar clinical cure (ceftriaxone 86.2% vs. cefazolin 90.2%, *p* = 0.35) [26]. Moreover, Winans et al. compared ceftriaxone to cefazolin in a variety of MSSA infections in OPAT settings and found similar favorable clinical outcomes (ceftriaxone 67.9% vs. cefazolin 79.8%, *p* = 0.17) [29]. Lastly, a systematic review by Kamfose et al. included six studies assessing ceftriaxone use for MSSA infections (only 1 study [22] evaluated MSSA BSIs), and they concluded that ceftriaxone was effective against MSSA infections [11]. Notably, an ongoing clinical trial is evaluating the efficacy and safety of ceftriaxone home therapy for MSSA infections and coagulate-negative staphylococcal infections [30]. Patients are randomly assigned to ceftriaxone or SOC (cloxacillin, cefazolin, or daptomycin) and evaluated at a 6-month follow-up.

Our findings expand the results of a recent meta-analysis by Yetmar et al. that compared ceftriaxone use with SOC in mixed MSSA infections [12]. Our findings were consistent with the meta-analysis by Yetmar et al., since they found that ceftriaxone was not different from SOC in the assessment of 90-day mortality (OR 0.93, 95% CI: 0.46–1.88) and hospital readmission (OR 0.96, 95% CI: 0.57–1.64). However, Yetmar et al. contradicted our finding of ADRs, since the authors found that ceftriaxone had statistically significantly lower ADRs (OR 0.49, 95% CI: 0.27–0.88). A likely explanation is that our meta-analysis included studies in both in-patient acute care and OPAT settings, while Yetmar et al. included studies only in OPAT settings with a variety of infections. Therefore, the difference in study design may explain the better tolerability of ADRs in the Yetmar et al. study as patients in OPAT care are generally more stable. Lastly, Yetmar et al. evaluated the recurrence of all MSSA infections (not only BSIs), and ceftriaxone was not different from SOC (OR 1.04, 95% CI: 0.63–1.72) [12]. Other methodological differences compared to our study were as follows: Yetmar et al. included studies in OPAT settings, excluded 6 studies published in conferences as abstracts, and did not evaluate clinical and microbiological cures. Overall, Yetmar et al. included seven studies: five studies [8,9,10,22,24] evaluated BSIs, one study [31] evaluated osteoarticular infections, and one study [29] evaluated mixed infections. Our systematic review and meta-analysis included 12 studies restricted to ceftriaxone use only for MSSA BSIs, and we evaluated six outcomes.

Since ceftriaxone exerts its pharmacodynamic action on bacteria by protein unbound form, concerns exist due to the high protein binding of ceftriaxone (85–95%) that may result in treatment failure [6,32]. However, the high protein binding of ceftriaxone does not appear to impact the clinical effectiveness based on our results from the meta-analysis. A recent in vitro study by Heffernan et al. evaluated ceftriaxone pharmacodynamics and optimal dosing regimens against MSSA isolates using an infection model [33]. The authors evaluated multiple ceftriaxone dosing regimens (1 g once daily, 2 g once daily, 1 g twice daily, and 2 g twice daily), and found that only high-dose ceftriaxone (2 g twice daily) achieved either sustained bacterial inhibition or killing in the first 24 h of therapy. Currently, the labeling information recommends using a ceftriaxone dose of 2 g twice daily only for patients treated for bacterial meningitis empirically with other appropriate antibiotics and patients treated for infective endocarditis due to enterococcal species in combination with ampicillin [6]. Therefore, Heffernan et al. concluded that ceftriaxone at the routinely used doses may not be a reliable choice, and alternative antibiotics should be used for MSSA infections [33]. The study by Heffernan et al. should be interpreted with caution since it is an in vitro study and the ceftriaxone dosing regimens have not been validated clinically. Furthermore, several studies [10,21] investigated predictors of treatment failure that may lead to mortality or hospital readmission and found that incomplete antibiotic course, lack of source control, metastatic foci of infections, heart failure, infective endocarditis, critical illness, and obesity are associated with MSSA BSI treatment failure.

In our analysis, a limited number of patients (11.5%) across seven studies [8,10,20,21,22,26,28] had infective endocarditis as a source of BSI. Five studies [8,9,10,22,26] reported which treatment patients with infective endocarditis received, with more patients receiving the SOC. Specifically, the treatment distribution for patients with infective endocarditis is as follows: Hamad et al. [9] (ceftriaxone 28.4% vs. oxacillin/cefazolin 41%, *p* < 0.02), Carr et al. [10] (ceftriaxone 6% vs. SOC 13.2%, *p* = 0.086), Patel et al. [22] (ceftriaxone 2.4% vs. nafcillin/cefazolin 13.7%, *p* < 0.07), Barber et al. [8] (ceftriaxone 0% vs. nafcillin/oxacillin/cefazolin 4.3%, *p* = 1), and Mohamed et al. [26] (ceftriaxone 4.3% vs. cefazolin 2.3%, *p* = 0.18). A plausible explanation is that patients with infective endocarditis as a source of BSI have high bacterial inoculum and should be treated based on the SOC for infective endocarditis [34]. Moreover, Hamad et al. found that the ceftriaxone arm had a lower rate of infective endocarditis (ceftriaxone 28.4% vs. oxacillin/cefazolin 43.2%, *p* = 0.02), valvular heart disease (ceftriaxone 13% vs. oxacillin/cefazolin 33.7%, *p* < 0.01), and transthoracic echocardiograms performed (ceftriaxone 25% vs. oxacillin/cefazolin 59%, *p* < 0.01) [9]. Therefore, patients with MSSA BSIs due to infective endocarditis were underrepresented in the ceftriaxone arms, and our findings do not support ceftriaxone use for MSSA BSIs with infective endocarditis as a source.

Notably, we were unable to perform some additional analyses because of the limited available data. Specifically, in addition to the ADRs analyzed above, *Clostridioides difficile* infection was reported by two studies [10,26], but it was inadequate to perform a meta-analysis. The rate of *C. difficile* infection was not statistically different between ceftriaxone and SOC: Carr et al. (ceftriaxone vs. SOC: 3% vs. 5.3%) and Mohamed et al. (ceftriaxone vs. SOC: 5.7% vs. 5%). We also attempted to evaluate MSSA BSI recurrence, but the number of studies was inadequate to perform a meta-analysis. The rate of infection recurrence was not statistically different between ceftriaxone and SOC: Carr et al. [10] (ceftriaxone vs. SOC: 12.1% vs. 5.3%), and Barber et al. [8] (ceftriaxone vs. SOC: 10% vs. 13%).

Regarding study limitations, all of the included studies are observational in nature, so the quality of data may be suboptimal, and there is a risk for bias. However, in our quality assessment using the NOS, 11 of the included studies were deemed of high quality and 1 was deemed of medium quality. There was also heterogeneity in the included studies due to different sources of MSSA BSIs. However, the random-effects model was used in the meta-analysis to adjust for the heterogeneity. Some studies did not report the complete regimen information for ceftriaxone or SOC. We also could not evaluate ceftriaxone efficacy based on patient age groups. Additionally, infectious disease consultation is associated with improved BSI treatment outcomes, reduced in-hospital mortality, and shorter hospitalization [35], but we could not evaluate the impact of infectious disease consultation due to a lack of data. Moreover, we could not evaluate the effectiveness of ceftriaxone based on the source of the MSSA BSI because only two studies [9,22] performed subgroup analyses to evaluate outcomes based on the source of MSSA BSI. Patients with infective endocarditis as a source of BSI were underrepresented in the ceftriaxone arms, and thus our findings do not support ceftriaxone use for those patients. Lastly, we included only studies published in the English language.

## 5. Conclusions

Ceftriaxone is a viable option for the treatment of MSSA BSIs in acute care and OPAT settings (except for patients with infective endocarditis as a source of BSI). Administration of ceftriaxone for MSSA BSIs is convenient because it is a once-daily antibiotic administered as a peripheral IV bolus injection. To our knowledge, the present study is the first systematic review and meta-analysis for ceftriaxone use specifically for the treatment of MSSA BSI. The study can inform clinicians that ceftriaxone was non-inferior to SOC for the treatment of MSSA BSI in terms of clinical cure, microbiological cure, mortality, hospital readmission, and ADRs. Clinicians should take into consideration patients’ factors that necessitate the dose adjustment of ceftriaxone or using an alternative agent to optimize therapy.

## Figures and Tables

**Figure 1 antibiotics-11-00375-f001:**
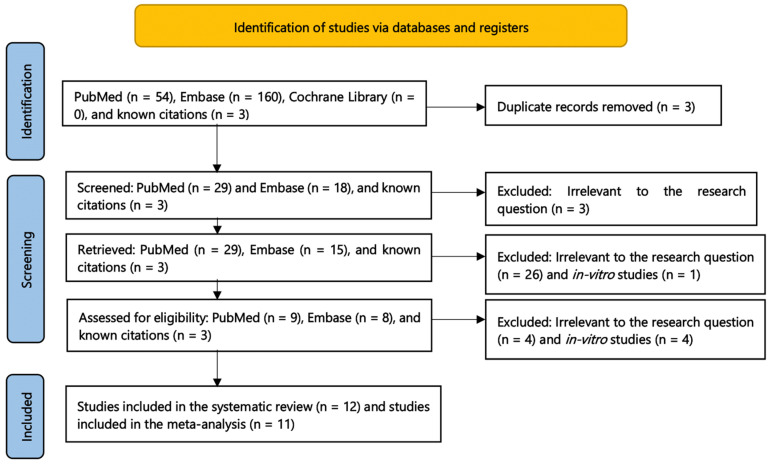
Flow diagram for Preferred Reporting Items for Systematic Reviews and Meta-Analyses (PRISMA).

**Figure 2 antibiotics-11-00375-f002:**
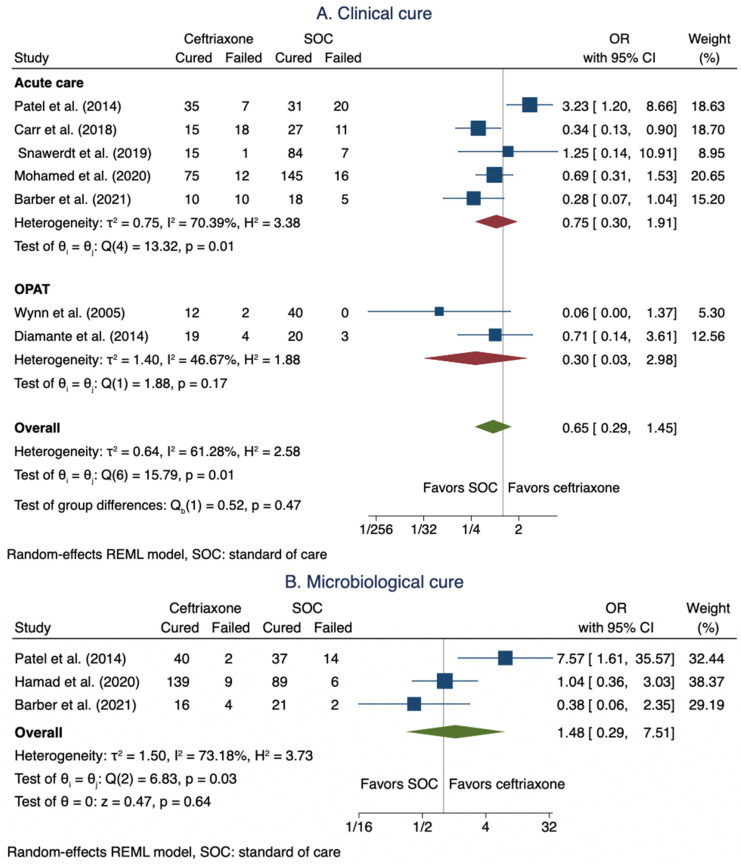
Meta-analysis of clinical cure (**A**) and microbiological cure (**B**).

**Figure 3 antibiotics-11-00375-f003:**
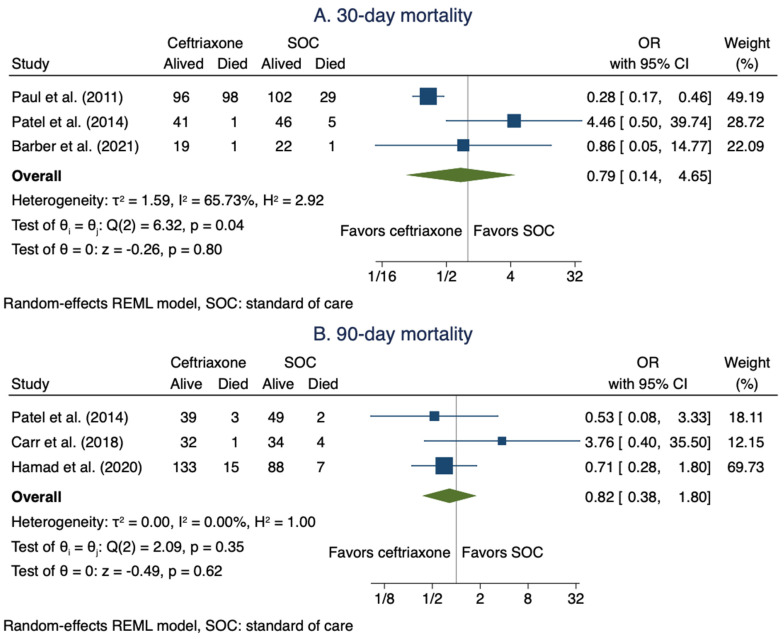
Meta-analysis of 30-day mortality (**A**) and 90-day mortality (**B**).

**Figure 4 antibiotics-11-00375-f004:**
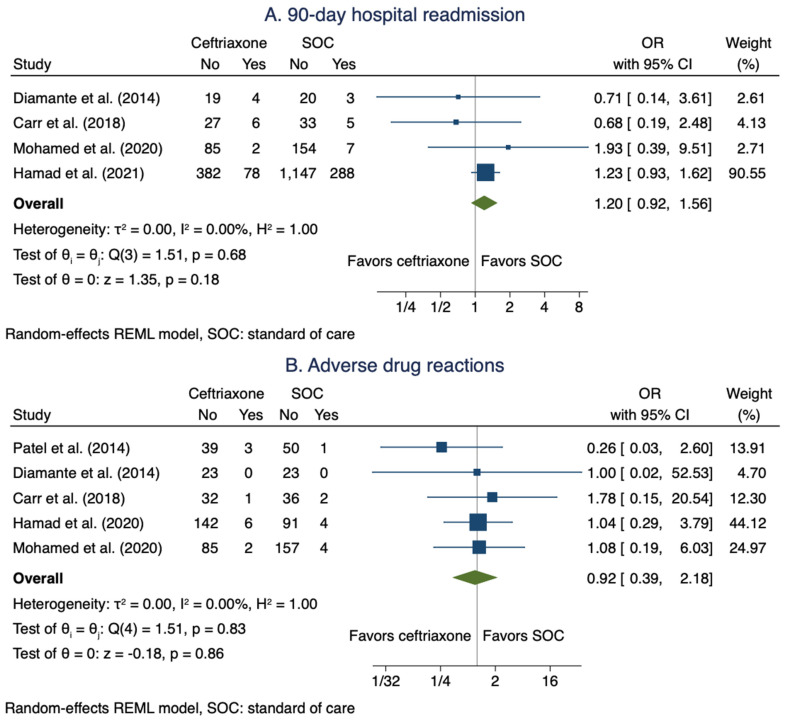
Meta-analysis of 90-day hospital readmission (**A**) and ADRs (**B**).

**Table 1 antibiotics-11-00375-t001:** Characteristics of the included studies.

Author	Region and Date of the Study	Age Distribution (Years)	Treatment Setting and Sample Size (N)	Suspected Sources of MSSA BSIs ^¥^	Ceftriaxone Regimens	SOC Regimens	NOS *
Paul et al. (2011) [20]	Israel, 1988–1994 and 1999–2007	Mean = 69 (SD: 16.8)	Rabin Medical Center, (N = 489)	Unknown = 147 (27.2%), SSTIs = 83 (15.3%), CRIs = 63 (11.6%), RTIs = 54 (10%), SSIs = 49 (9.1%), other endovascular infections = 41 (7.6%), infective endocarditis = 25 (6.5%), and OAIs = 27 (5%)	Ceftriaxone and cefotaxime (ceftriaxone = 176 and cefotaxime = 18) ^±^	Cloxacillin or cefazolin	9
Patel et al. (2014) [22]	U.S., January 2000–September 2009	Ceftriaxone: mean = 63 (SD: 12.6) and SOC: mean = 68 (SD: 12.5)	Edward Hines, Jr. VA Hospital, (N = 93)	Unknown = 22 (23.6%), OAIs = 22 (23.6%), CRIs = 20 (21.5%), SSTIs = 15 (16.1%), infective endocarditis = 8 (8.6%), UTIs = 4 (4.3%), and RTIs = 2 (2.1%)	Ceftriaxone 2g Q 24 hr for 14 days for uncomplicated BSIs, and 28 days for complicated BSIs ^¶^	Nafcillin or cefazolin for 14 days for uncomplicated BSIs, and 28 days for complicated BSIs ^§¶^	9
Carr et al. (2018) [10]	U.S., January 2009–August 2014	Ceftriaxone: mean = 64 (SD: 13.6) and SOC: mean = 63 (SD: 10.7)	Louis Stokes Cleveland Department of VA Medical Center, (N = 71)	OAIs = 28 (39.4%), endovascular infections = 17 (23.9%), SSTIs = 14 (19.7%), unknown = 9 (12.6%), infective endocarditis = 7 (9.9%), and UTIs = 3 (4.2%)	Ceftriaxone for 14 days ^¶^	Cefazolin for 14 days ^¶^	8
Hamad et al. (2020) [9]	U.S., December 1, 2014–April 30, 2019	Median = 59.6 (IQR: 47.8–70)	Discharged from Barnes-Jewish Hospital on OPAT, (N = 243)	Infective endocarditis = 83 (34.2%), CRIs = 70 (28.8%), OAIs = 68 (28%), unknown = 40 (16.5%), SSTIs = 33 (13.6%), prosthetic material infections = 26 (10.7%), SSIs = 16 (6.6%), and CNS = 13 (5.4%)	Ceftriaxone 2–4 g Q 24 hr for at least 7 days	Oxacillin 2g Q 4 hr or cefazolin 2g Q 8 hr for at least 7 days	9
Barber et al. (2021) [8]	U.S., February 1, 2015–January 21, 2016	Ceftriaxone: median = 43.5 (IQR: 35.2–57.5) and SOC: median = 45 (IQR: 36–55)	University of Mississippi Medical Center, (N = 43)	OAIs = 11 (25.6%), CRIs = 9 (20.9%), SSTIs = 7 (16.3%), unknown = 4 (9.3%), RTIs = 3 (7%), SSIs = 2 (4.7%), CNS = 1 (2.3%), and infective endocarditis = 1 (2.3%)	Ceftriaxone for at least 2 days ^¶^	Nafcillin, oxacillin, or cefazolin for at least 2 days	8
Snawerdt et al. (2019) [23]	U.S., February 2016–February 2018	NA	Multi-centers, (N = 222, 107 patients with BSIs)	NA	Ceftriaxone ^¶^	Cefazolin or nafcillin ^¶^	7
Diamante et al. (2014) [28]	U.S., January 2011–December 2013	NA	Parkland Hospital OPAT clinic, (N = 46)	OAIs = 26 (56%), SSTIs = 7 (15%), CRIs = 7 (15%), and infective endocarditis = 6 (13%)	Ceftriaxone ^¶^	Cefazolin ^¶^	7
Hamad et al. (2021) [21]	U.S., 2010–2018	NA	Barnes-Jewish Hospital OPAT, (N = 1895)	SSTIs = 757 (40%), OAIs = 745 (39.3%), SSIs = 558 (29.4%), RTIs = 356 (18.8%), infective endocarditis = 276 (14.6%), CNS = 200 (10.6%), device related infections = 192 (10.1%), and CRIs = 175 (9.2%)	Ceftriaxone ^¶^	Cefazolin or oxacillin ^¶^	9
Wynn et al. (2005) [24]	U.S., 1996–August 2001	NA	OPAT registry, (N = 1252; 54 patients with BSIs)	NA	Ceftriaxone 1–6 g/day ^¶^	Cefazolin 1.5–12 g/day, oxacillin 2–48 g/day, or nafcillin 0.8–24 g/day ^€¶^	7
Falsetta et al. (2017) [25]	U.S., January 2012–September 2016	NA	Acute care, (N = 51)	NA	Ceftriaxone for at least 14 days ^¶^	Cefazolin or nafcillin for at least 14 days ^¶^	7
Mohamed et al. (2020) [26]	U.S.	Ceftriaxone: mean = 57.4 (SD: 16.8) and SOC: mean = 61 (SD: 15.9)	Saint Luke’s Health System, (N = 248)	Unknown = 75 (30.2%), OAIs = 53 (21.4%), SSTIs = 51 (20.6%), RTIs = 21 (8.5%), device related infections = 11 (4.4%), CNS = 11 (4.4%), infective endocarditis = 9 (3.6%), and UTIs = 4 (1.6%)	Ceftriaxone ^¶^	Cefazolin ^¶^	7
Bhavan et al. (2018) [27]	U.S.	NA	Parkland Hospital OPAT clinic, (N = 258, 135 patients with BSIs)	NA	Ceftriaxone ^¶^	Cefazolin ^¶^	5

Abbreviations: BSIs: bloodstream infections, CNS: central nervous system, CRIs: catheter-related infections, g: gram, hr: hour, IQR: interquartile range, NA: not available, No.: number, NOS: Newcastle Ottawa scale, OAIs: osteoarticular infections, OPAT: outpatient parenteral antibiotic therapy, Q: every, RTIs: respiratory tract infections, SD: standard deviation, SSTIs: skin and soft tissue infections, SSIs: surgical site infections, UTIs: urinary tract infections, VA: Veterans Affairs, ^¥^: some patients have multiple suspected sources of BSIs, *: score out of 9 (1–3 low quality, 4–6 medium quality, 7–9 high quality), and ^¶^: complete regimen information was not reported, ^±^: we included this study because cefotaxime is similar to ceftriaxone in the microbiological spectrum and only comprise 9.2% of the ceftriaxone group, ^§^: the SOC arm only included nafcillin and cefazolin and we excluded vancomycin, ^€^: the SOC arm only included oxacillin, nafcillin, and cefazolin. Data on other treatments such as vancomycin and clindamycin were excluded.

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
