# Peer review of "Effectiveness and Safety of Ceftriaxone Compared to Standard of Care for Treatment of Bloodstream Infections Due to Methicillin-Susceptible Staphylococcus aureus: A Systematic Review and Meta-Analysis"

_antibiotics, 2022, doi:10.3390/antibiotics11030375_

Round 1

Reviewer 1 Report

This manuscript titled as “Effectiveness and safety of ceftriaxone compared to standard of care for treatment of blood stream infections due to methicillin susceptible Staphylococcus aureus: A systematic review and meta-analysis” attempted to assess ceftriaxone efficacy against MSSA focusing on BSIs performing a systematic review of previously reported studies. As the authors mentioned, ceftriaxone could be convenient to use for both patients and clinicians due to its relatively simple regimen.  Therefore, it would be worth investigating its clinical efficacy in a detailed and systematic manner.  Although the analysis methods in this study are robust, there are a few comments that the authors could consider to make the study more comprehensive.  The detailed comments are as follows.

Age is an important factor that could affect drug treatment outcomes but the authors only mentioned “adult patients” without specifying the age distribution of the patients subjected in the study. It would be very informative to analyze ceftriaxone efficacy based on patients’ ages.  Among 12 studies subjected in the authors’ analyses, 11 studies were done in US. Since there might be a regional difference, it would be better to carefully compare all parameters in the exceptional case (Paul et al, 2010) with other studies, and also mention that 11 studies are based on US data but not the one case. 

Several minor points are as follows.

Line 41-42, it would be better to have a more recent reference.

Figure 1, “Duplicate records removed (n=3)”. Is this correct?

Table 1, the region information is absent for the data set of Snawerdt et al (2019) and Falsetta et al (2017)

Some of the references’ published years do not match in Table 1, Figure 2, Figure 3, Figure 4, and References.

Reviewer 2 Report

Thank you for your submission. I enjoyed reading your SR and MA. 

In your discussion or limitation section please address the lack of standardization of doses across the studies. The studies used had doses of ceftriaxone optimized, but no data is available for the comparator groups. This is a significant flaw in the studies that should be addressed. Studies reference # 6, 8, 18, 19, 20, 21, 23, 24, 25 did not include doses for the comparator group. We are unable to determine if the doses were optimized based on renal function and may not represent a true comparison to ceftriaxone. 

References #22 and #20 included vancomycin or vancomycin and clindamycin as the standard of care.  This should be addressed as a limitation in that the standard of care includes non-beta-lactams. From previous studies including the one from Stryjewski ME etal in 2007 (PMID: 17173215, DOI:10.1086/510386), we know that vancomycin is not as robust as a beta-lactam for MSSA bacteremia. 

Lastly, the studies included in the MA did not discuss other external factors that may impact mortality such as an infectious disease consultation (PMID: 25701854, DOI: 10.1093/cid/civ120). I think this should be included in the study limitations as well. 

Reviewer 3 Report

Comments:

In this Ms, the authors describe the effectiveness and safety of ceftriaxone as a systematic review and a meta-analysis on twelve retrospective cohort studies.

Ms. exhibits a high scientific standard and quality. The content is appropriately reported in well-ordered headings and subheadings. The final part of the discussion section is dealing with study limitations, which are not based on the present Ms. but depend on significant limitations of other studies in this field.

There are several issues about this Ms. that need modification.

Introduction

In the introduction section, it would be very useful to add a very updated reference (Jin Y. Et al., 2022. Emerging Microbes Infections 11:326-336)

Line 46: “…and for 4-6 weeks….” please change as follows “… and up to 4-6 weeks…”

Line 51: Please change “…..push…” to “…bolus injection….”

Conclusion

Line 302: Please change “…..push…” to “…bolus injection….”
